

# Defense mechanism responses to COVID-19

Nouf Altwaijri[1], Turki Abualait[2], Mohammed Aljumaan[3], Raidah Albaradie[4], Zahid Arain[5] and Shahid Bashir[4]

[1] King Saud University, Riyadh, Saudi Arabia
[2] College of Applied Medical Sciences, Imam Abdulrahman Bin Faisal University, Dammam, Saudi Arabia
[3] College of Medicine, Imam Abdulrahman Bin Faisal University, Dammam, Saudi Arabia
[4] Neuroscience Center, King Fahad Specialist Hospital, Dammam, Saudi Arabia
[5] Liver Transplant Department, King Fahad Specialist Hospital, Dammam, Saudi Arabia

## ABSTRACT

The COVID-19 pandemic has had a wide range of negative physical and mental impacts. This review begins with a theoretical explanation of the psychological defense mechanisms used to deal with the pandemic. It then discusses different categories of defense mechanisms and their roles in managing the impacts of psychological distress. The aim of this review is to highlight the various psychological defense mechanisms individuals use to deal with the pandemic and to discuss how adjustment mechanisms can protect individuals from internal and external threats by shielding the integrity of the ego (the mind) and helping individuals maintain their self-schema.

## INTRODUCTION

The COVID-19 pandemic has had serious negative impacts on the physical and mental health of individuals around the world (*Holmes et al., 2020*). Several reports, on the general population, have shown that the psychological adverse effects of contagion and government-enforced quarantines, during the pandemic, extends afar the fear of viral infection and comprises social separation impacts, feeling helpless, loss of freedom, and uncertainty of disease progression (*Li et al., 2020*; *Cao et al., 2020*). Such effects are associated with internalizing (anxious, depressive symptoms, social withdraws, and somatic complaints) and externalizing (oppositional, aggressive and violent/delinquent) psychological behavior that might be developed intensely (*Carragher et al., 2015*). These psychological behavior can be evolved dramatically into life-threatening consequences such as suicide ideation and action (*De Berardis et al., 2018*).

Furthermore, evidences clearly have shown that frontline healthcare workers, who were closely involved in diagnosing, medically managing and treating infected patients with COVID-19 virus, in addition to witnessing the suffering and death of patients during the pandemic were highly likely to evolve psychological distress and mental health impairments (*Thombs et al., 2020*; *Olashore et al., 2021*; *Bahar Moni et al., 2021*). These impairments including a wide range of non-specific symptoms of anxiety, distress, fear, depression, and insomnia, might ultimately trigger long-lasting and clinically

Corresponding author
Shahid Bashir,
shahidbpk13@gmail.com

significant personal and psychiatric disorders (*Xiao et al., 2020*; *Lin, 2020*; *Labrague & Santos, 2021*; *Rajabimajd, Alimoradi & Griffiths, 2021*).

Given the fact that the COVID-19 pandemic has negative influences on psychological well-being, it is crucial to further explore the human behaviors that are usually elicited by the psychological distress. The pandemic's effects reach beyond the viral infection and consist of complex psychodynamic processes that continuously impact individuals' behavior, emotions, and life satisfaction (*Brooks et al., 2020*).

In behavioral science, defense mechanisms have demonstrated their usefulness to practitioners and theoreticians. The concept of defense is vague as each individual views it differently. However, the number of recognized defense mechanisms has been increasing over the years, and their application has been expanding (*Gabbard, 1995*). In general, defense mechanisms are unconscious psychological strategies individuals use to protect themselves from the anxiety that arises from unacceptable feelings or thoughts (*Bailey & Pico, 2021*). They include denial, projection, isolation, intellectualization, rationalization, displacement, sublimation and reaction formation (*Cramer, 2015*; *Thomä, 1995*; *Freud, 1966*). Since the inclusion of the structural model of personality (id, ego and superego) in psychoanalytic theory, the concept of defense has been regarded as a mental function and a part of the ego (*Cramer, 2015*; *Freud, 1936*; *Frederickson, Messina & Grecucci, 2018*). Defense mechanisms are significant predictors of several variables, including personality traits, identity status, psychopathology, and the development of the ego (*Cramer, 2018*; *Sullivan, 2013*). Without them, the conscious mind would be more vulnerable to negatively charged emotional input (*Busch et al., 2011*; *Eley & Stevenson, 2000*; *Finlay-Jones & Brown, 1981*; *Mathews & Klug, 1993*; *Rapee, 1997*; *Shaver et al., 1987*). Valliant proposes the existence of four types of defenses: psychotic defenses, immature defenses, neurotic defenses, and mature defenses (*Cramer, 2015*; *Vaillant, 1992*). According to psychoanalytic theory, defense mechanisms are, in short, unconscious strategies aimed to decrease or eliminate anxiety due to potentially harmful or unacceptable stimuli; they may or may not be associated with a specific behavior (*Vaillant, 2011*; *Walker & McCabe, 2021*). After a physical injury, redness, tenderness, and swelling immobilize a fracture so it can heal. Similarly, to physical illnesses, the symptoms of mental illnesses, such as depression or anxiety, are a product of homeostatic struggles with adapting to life (*Zechowski, 2017*).

Any individually perceived stress can provoke psychological defense mechanisms (*Vaillant, 2011*; *Walker & McCabe, 2021*). Perceived stress and tension resulted from government-enforced quarantines and lockdowns, social distancing, and other preventive measures associated with the COVID-19 pandemic can provoke psychological defense mechanisms (*Marčinko et al., 2020*). According to psychoanalytic theory, a defense mechanism is an unconscious or automatic psychological strategy that aims to lessen anxiety arising from intolerable or potentially threatening impulses (*Vaillant, 2011*). It has been theorized that the defense mechanisms employed in response to the COVID-19 pandemic are more likely to be immature, as anxiety over the pandemic can be temporarily relieved by altering painful ideas or distorting one's external reality (*Walker & McCabe, 2021*; *Marčinko et al., 2020*). For example, disavowal is a common defense mechanisms that can be used to protect an individual from something they cannot cope with

(*Marčinko et al., 2020*). Usually, stressors restore internal homeostasis by generating adaptive behavioral and physiological responses (*Cooper, Clinard & Morrison, 2015*). However, when those stressors are uncontrollable, severe, or prolonged, they can lead to negative health consequences, such as post-traumatic stress disorder (*Cooper, Clinard & Morrison, 2015*; *Abelson et al., 2007*; *Meewisse et al., 2007*; *Heim et al., 2008*). The underpinning substrates of defense mechanisms include a collection of mental processes that manipulate or in some other way reject reality to protect an individual from overwhelming feelings of stress and anxiety and from negative stimuli. This can lead to a balance between subjectivity and reality (*Vaillant, 2011*; *American Psychiatric Association, 2013*). This kind of mechanism protects against internally generated threatening wishes and desires and from external, potentially harmful threats. It can therefore shield the integrity of the ego (mind) and maintain an individual's self-schema (*Vaillant, 2011*). Adjustment after trauma includes resilience and coping with conscious and unconscious content (*Marčinko et al., 2020*).

While living through this worldwide health crisis, individuals have employed a variety of explicit and implicit emotional coping strategies to alleviate the impact of pandemic-related psychological distress (*Gyurak, Gross & Etkin, 2011*; *Rice & Hoffman, 2014*). These defense mechanisms might lead to coping and adjustment (healthy) behavior or unhealthy (pathological) behavior, which can have serious sequelae. Whether a defense mechanism leads to healthy or unhealthy behavior depends on the environment in which the mechanism is utilized (*Rice & Hoffman, 2014*). Individuals' responses to stressful situations might fall under one of the four categories of defense mechanisms; these responses can improve resilience and protect the ego from internal conflicts (*Vaillant, 2011*). This article serves as a narrative review has two emphases of interest: (1) the assessment of the negative influences of the COVID-19 pandemic on the psychological well-being and how the psychological distress, due to the pandemic, would affect and manipulate human's psychological behavior; and (2) the importance of investigating different psychological defense mechanisms that play a critical role in mitigating psychological distress and utilizing mechanisms that promotes mental resilience, in addition, the evaluation of how adjustment mechanisms and coping strategies can protect peoples from internal and external threat.

## REVIEW METHODOLOGY

Literature review was done on PubMed on the relationship of COVID-19 pandemic and defense mechanism, using the keywords "COVID-19" OR "pandemic"; "COVID-19 pandemic", AND "Defense mechanism" OR "Self-mechanism". It also examines evidence of the role of adaptive defense mechanisms in protecting people from psychological distress due to stressful life events, such as the COVID-19 pandemic.

### Pathologic defenses

Pathologic defenses, also called psychotic defenses, are mainly characterized by an evident break with objective reality (*Zechowski, 2017*). They permit the individual to restrict conflicts to the external world. In other words, they involve reorganizing external experiences to reduce the need to adjust to reality. Individuals who use these mechanisms

may appear irrational or psychotic to others; these defenses represent the most dysregulated level of adjustment and are characterized by a failure to contain the defensive reaction to stressors, leading to a break with objective reality (*Vaillant, 1971*; *Berney et al., 2014*). Pathologic defenses are common in individuals with post-traumatic (stress disorder (PTSD))  and may include delusional projections and psychotic distortion or denial. These mechanisms are common in three situations: in dreams, in cases of psychosis, and among young children. To breach them, neuroleptics, or waking the dreamer, are needed to alter the brain (*Vaillant, 2011*). Psychotic defense mechanisms are an issue in the study of personality and of schizophrenic and bipolar disorders (*Berney et al., 2014*).

## Immature defenses

Immature defenses mainly comfort the user; however, they annoy the observer (*Zechowski, 2017*). These defenses reduce feelings of anxiety and distress generated by perceived harmful impulses or an unacceptable reality. Individuals using such defenses may display outbursts of behavior or emotions that may be socially inappropriate. The goal of such outbursts is to mask undesirable underlying feelings or thoughts. The twelve types of immature defenses are projection, isolation of affect, dissociation, splitting, rationalization, devaluation, denial, acting out, autistic fantasies, somatization, passive-aggression, and displacement. They are used to suppress emotional awareness and can contribute to psychopathology (*Costa & Brody, 2013*). These kinds of defenses are common among adolescents, substance abusers, and individuals with personality disorders or brain injury. Those defenses rarely respond to verbal interpretations on their own (*Vaillant, 2011*).

Immature defenses are a factor in treating patients with depression. One study found that tracking the use of such defenses during a two-year follow-up period led to a significant decrease in the need for treatment compared to a control group (*Bond, 2004*; *Akkerman, Lewin & Carr, 1999*). Another paper notes that in patients who experienced symptomatic recovery, the use of immature defenses declined significantly, while no change was observed in neurotic or mature defenses (*Akkerman, Carr & Lewin, 1992*). The most common immature defense in individuals with depression is projection (*Bond, 2004*; *Spinhoven & Kooiman, 1997*). Immature defenses can be breached in several ways. The first is confrontation, which is usually done by a group of supportive peers or in focused psychotherapy (*Vaillant, 2011*; *Michal & Osborn, 2021*). Therefore, during the COVID-19 pandemic and other difficult times, it is vital to stay connected to support systems. The second approach involves rendering the individual less anxious or lonely by improving intrapsychic competence through empathy. The third approach is to improve brain function, such as by relieving normal-pressure hydrocephalus (*Vaillant, 2011*; *Perry et al., 2008*; *Damasio, 1999*).

## Neurotic defenses

Neurotic defenses keep all potentially threatening feelings, memories, ideas, wishes, or fears outside an individual's awareness (*Vaillant, 2011*). They can be used to manage or control an external violent and vulnerable environment and to avoid underlying feelings of anxiety and guilt. This kind of defense offers short-term rewards for dealing with stressful

events; however, long-term use of such defenses can lead to a wide range of relationship challenges. These defenses manifest clinically as compulsions, somatizations, phobias, and amnesia. Unlike immature defenses (*Vaillant, 2011*), neurotic defenses typically make the user more uncomfortable than the observer (*Vaillant, 2011*; *Zechowski, 2017*). Neurotic defenses can commonly be breached with psychotherapy (*Vaillant, 2011*).

Evidence indicates that neurotic defenses play a critical role in mediating the psychological effects of the COVID-19 pandemic. Several dynamic coping strategies that fall into this category have been employed to buffer individuals from the effects of the pandemic. One example is repression. This may involve avoiding conscious awareness of or exposure to information about the dangers of COVID-19. However, this knowledge continues to influence peoples' behavior; it does not simply vanish. Another example is dissociation, such as when healthcare workers try to dissociate or disconnect from negative experiences, memories, thoughts, sensations, and even their identities during the pandemic. A third example is reaction formation, which is when individuals express the opposite of their real feelings and emotions. This kind of transformation of emotions may help promote mental resilience. Another example of this kind of defense mechanism is displacement, such as when a patient infected with COVID-19 expresses anger and blames family members or others for their illness.

### Mature defenses

Mature defense are conscious processes that boost feelings of control and augment positive emotions; they result in optimal adaptation to stressors (*Zechowski, 2017*). These defenses typically enhance gratification and increase one's conscious awareness of their own ideas and feelings and their consequences (*Vaillant, 2011*; *Zechowski, 2017*). They can also help individuals balance conflicting motivations (*Cramer, 2018*). Examples of mature defenses include altruism, sublimation, suppression, and humor. Even though humor seems to reflect dissociation and denial, it, like meditation, can help shift the body to parasympathetic calmness from autonomic sympathetic agitation (*Vaillant, 2011*). Mature defenses increase with age (*Vaillant, 2011*; *Stellar, 2021*), in contrast to PTSD, which usually decreases in severity with age (*Vaillant, 2011*). Coping responses to stress can be broadly divided into three categories: (1) voluntarily asking for help from appropriate persons (such as mobilizing social supports), (2) voluntary strategies (such as gathering information and anticipating danger) (*Vaillant, 2011*; *Tovote, Fadok & Lüthi, 2015*), and (3) unconscious homeostatic mechanisms such as fever or leukocytosis; these are involuntary physical responses that help reduce the effects of stress (*Vaillant, 2011*).

## ADAPTIVE DEFENSE MECHANISMS AND THE COVID-19 PANDEMIC

Research has demonstrated the role of adaptive defense mechanisms in protecting people from psychological distress due to stressful life events, such as the COVID-19 pandemic. Defense mechanisms have played an important role in mitigating psychological distress during the COVID-19 pandemic, and utilizing defense mechanisms promotes mental resilience (*Walker & McCabe, 2021*; *Di Giuseppe et al., 2020c*). Walker et al. identified

several psychological defense mechanisms employed during the COVID-19 pandemic, including denial, hypochondriasis, altruism, sublimation, and humor. Of these, altruism, sublimation, and humour are highly adaptive defenses that maximize gratification and help individuals become more aware of their own feelings and ideas and their consequences (*Walker & McCabe, 2021*).

It is important to consider defense mechanisms within the larger structure of stress and coping (*Zechowski, 2017*). A recent meta-analysis of stressful and traumatic experiences during the COVID-19 pandemic found high rates of anxiety (31.9%), depression (33.7%), and stress (29.6%) (*Salari et al., 2020*). Such psychological impacts can lead to various acute anxiety reactions, which may be accompanied by maladaptive behavior or immature defense mechanisms. The pandemic and lockdowns have impacted young children and adolescents more significantly than adults due to their immature emotional and social development. Younger children are more likely to experience clinginess and the fear of COVID-19 affecting family members; in older children, impacts may take the form of inattention or constant inquiries about COVID-19. Increased irritability, inattention, and clinginess have been observed in children of all age groups (*Singh et al., 2020*; *Viner et al., 2020*).

During any global pandemic, the use of defense mechanisms should be continuously observed since the capability to properly utilize a variety of defenses in perplexing environments is associated with mental and psychological health. Furthermore, the use of immature defenses is a pathogenic predictor for psychiatric conditions (*Prout et al., 2020*). Distress increases when individuals rely less on adaptive defense mechanisms (*Prout et al., 2020*).

*Di Giuseppe et al. (2020c)* note that the overall defensive functioning of participants during the first week of lockdown in the ongoing pandemic fell within normal neurotic ranges. However, in individuals experiencing distress or depression, overall defensive functioning correlated negatively with the severity of symptoms (*Di Giuseppe et al., 2020c*). The use of immature defenses correlated positively with more severe symptoms (*Di Giuseppe et al., 2020c*). Moreover, higher defensive functioning (of mature defenses) correlated negatively with levels of depression and post-traumatic stress (*Walker & McCabe, 2021*; *Di Giuseppe et al., 2020c*). However, when immature defense mechanisms such as neurotic behavior are used, the risk of developing pathological personality traits increases (*Carvalho, Reis & Pianowski, 2019*). Despite this, neurotic defenses play a role in managing virus-related anxiety (*Walker & McCabe, 2021*; *Prout et al., 2020*; *Di Giuseppe et al., 2020b*). Evidence indicates that neurotic defenses play a critical role in mediating the psychological effects of the COVID-19 pandemic. Several dynamic coping strategies that fall into this category have been employed to buffer individuals from the effects of the pandemic. One example is repression. This may involve avoiding conscious awareness of or exposure to information about the dangers of COVID-19. However, this knowledge continues to influence peoples' behavior; it does not simply vanish. Another example is dissociation, such as when healthcare workers try to dissociate or disconnect from negative experiences, memories, thoughts, sensations, and even their identities during the pandemic. A third example is reaction formation, which is when individuals express

the opposite of their real feelings and emotions. This kind of transformation of emotions may help promote mental resilience. Another example of this kind of defense mechanism is displacement, such as when a patient infected with COVID-19 expresses anger and blames family members or others for their illness.

The most common depressive symptoms during the pandemic include depression, anxiety, and PTSD (*Di Giuseppe et al., 2020c*; *Perry & Bond, 2012*; *Babl et al., 2019*). In an Italian cohort, different levels of defenses were related to varying degrees of adaptation to stress (*Di Giuseppe et al., 2020c*). In addition, emotional regulation is of utmost importance during stressful times, particularly among vulnerable groups, such as patients suffering from physical or mental illnesses (*Di Giuseppe et al., 2020c*; *Zilcha-Mano, 2021*; *Boldrini et al., 2020*; *Di Giuseppe et al., 2020*; *Martino et al., 2020a*; *Martino et al., 2020*; *Nam et al., 2019*). One study of defense mechanisms and coping strategies found that coping mechanisms, a positive attitude, and mature defense mechanisms negatively correlated with perceived stress (*Gori, Topino & Di Fabio, 2020*). In addition, life satisfaction, or the perception of having a full, meaningful life (*Gori, Topino & Di Fabio, 2020*; *Grecucci et al., 2015*), is associated with lower stress levels and more active adaptive coping (*Gori, Topino & Di Fabio, 2020*; *Hooker, Masters & Park, 2018*).

All these findings can be applied to responses to the COVID-19 pandemic. During this crisis, people have sought help from experienced personnel, prepared by complying with precautionary measures, and tried to fight COVID-19 if they became infected. Higher defensive functioning is associated with lower levels of depression, anxiety, and post-traumatic stress symptoms. On the other hand, as stated above, Italians with poorer adaptive defense functioning experienced higher levels of stress during the first week of lockdown (*Di Giuseppe et al., 2020c*).

## RECOMMENDATIONS

Individuals can take some steps to enhance their psychological defensive responses to COVID-19 or any similar pandemic. For example, physical exercise modulates the immune system; cytokines, both pro- and anti-inflammatory, are released during and after an exercise session. Exercise also increases cell recruitment and lymphocyte circulation (*Da Silveira et al., 2021*). Multiple studies have found that regular moderate exercising helps reduce the frequency of upper respiratory tract infections (URTIs) (*Larenas-Linnemann et al., 2020*; *Matthews et al., 2002*; *Fondell et al., 2011*; *Nieman et al., 2011*). This is because exercise helps stimulate cellular immunity (*Da Silveira et al., 2021*; *Pedersen & Hoffman-Goetz, 2000*; *Bajpai & Nahrendorf, 2021*). Adequate sleep patterns also help modulate the adaptive immune response (*Larenas-Linnemann et al., 2020*; *Spiegel, 2002*). Therefore, during the pandemic, it is vital to maintain an overall healthy lifestyle while also adhering to all recommended precautionary measures to decrease one's chances of becoming infected.

Seeking help when needed is also extremely important. Thanks to telemedicine, help is now within nearly everyone's reach. Although there are many challenges to telemedicine, during the pandemic, its positive features outweigh the negative. Moreover, offering continuous support within communities and families is of utmost importance during such

difficult times. Finally, individuals must cope with their current circumstances in healthy and productive ways by developing a more conscious relationship with reality.

## CONCLUSION

The COVID-19 pandemic has had a wide range of negative physical and mental impacts. This review has highlighted some of the psychological defence mechanisms individuals use to deal with the pandemic. Defense mechanisms are unconscious strategies utilized to protect individuals from anxiety or stress arising from intolerable stimuli. They are a collection of mental processes that defend against overwhelming feeling of stress and threatening stimuli and promote mental resilience. Individuals' responses to stressful situations might fall under one of the four types of defense mechanisms (pathologic, immature, neurotic, and mature); these defense mechanisms can improve resilience and protect the ego from internal conflict.

Individuals who employ higher levels of defensive functioning, such as mature defenses, are less likely to develop psychiatric problems or maladaptive behaviors. Individuals who use immature defense mechanisms, such as neurotic defenses, have a higher risk of developing pathological personality traits.

Individuals can take steps to enhance their own defense mechanism responses to COVID-19 or any similar pandemic. For example, physical activity modulates the immune system. Psychosocial interventions play a critical role in reducing the risk of developing psychiatric and personality disorders due to a crisis or pandemic. Individuals employ a wide variety of explicit and implicit emotional coping strategies to reduce the psychologically distressing impacts of the pandemic (*Rice & Hoffman, 2014*). The healthy, productive use of defense mechanisms can encourage coping and healthy behavior that helps counter psychological distress.

### Funding
The authors received no funding for this work.

### Competing Interests
The authors declare there are no competing interests.

### Author Contributions

- Nouf Altwaijri and Shahid Bashir conceived and designed the experiments, performed the experiments, analyzed the data, authored or reviewed drafts of the paper, and approved the final draft.
- Turki Abualait and Raidah Albaradie conceived and designed the experiments, performed the experiments, authored or reviewed drafts of the paper, and approved the final draft.
- Mohammed Aljumaan and Zahid Arain conceived and designed the experiments, authored or reviewed drafts of the paper, and approved the final draft.
## Data Availability

There is no raw data; this is a literature review.

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
