# Peer review of "Defense mechanism responses to COVID-19"

_PeerJ, doi:10.7717/peerj.12811_

## Round 0.1 · original submission · Major Revisions

Despite addressing a topic of utmost importance in these times, the quality and content of the manuscript is not enough to be accepted for publication in PeerJ in its current form. The overall content of the manuscript is too general and not contributing sufficient to the literature. Please see the reviewers comments and revise accordingly.

Furthermore, if the reviewer has requested to incorporate any references, please ignore this suggestion made by the reviewer, until or unless the requested reference is directly relevant to the work and will contribute to the manuscript effectively.

Reviewer 1 ·

Basic reporting

The study structure is confusing as I cannot know whether the authors want to report a narrative review or a systematic review. Moreover, the paper should be strengthened using the following comments.
1. The first paragraph in the Introduction should be strengthened using the following references to talk about the psychological distress during COVID-19 pandemic across different populations. Specifically, with the use of the evidence shown below, the authors can justify that the COVID-19 increases the psychological distress in different aspects for human being. Therefore, it is important to further investigate the behaviors that are usually triggered by the psychological distress.
Hasannia E, Mohammadzadeh F, Tavakolizadeh M, Davoudian N, Bay M. Assessment of the anxiety level and trust in information resources among iranian health-care workers during the pandemic of coronavirus disease 2019. Asian J Soc Health Behav 2021;4:163-8
Patel BR, Khanpara BG, Mehta PI, Patel KD, Marvania NP. Evaluation of perceived social stigma and burnout, among health-care workers working in covid-19 designated hospital of India: A cross-sectional study. Asian J Soc Health Behav 2021;4:156-62
Olashore AA, Akanni OO, Fela-Thomas AL, Khutsafalo K. The psychological impact of COVID-19 on health-care workers in African Countries: A systematic review. Asian J Soc Health Behav 2021;4:85-97
Sharma R, Bansal P, Chhabra M, Bansal C, Arora M. Severe acute respiratory syndrome coronavirus-2-associated perceived stress and anxiety among indian medical students: A cross-sectional study. Asian J Soc Health Behav 2021;4:98-104
Rajabimajd N, Alimoradi Z, Griffiths MD. Impact of COVID-19-related fear and anxiety on job attributes: A systematic review. Asian J Soc Health Behav 2021;4:51-5
Patil ST, Datar MC, Shetty JV, Naphade NM. “Psychological consequences and coping strategies of patients undergoing treatment for COVID-19 at a tertiary care hospital”: A qualitative study. Asian J Soc Health Behav 2021;4:62-8
2. The first paragraph in the Introduction should also mention different behaviors (e.g., vaccine uptake, protection behaviors) have been discussed in the literature (please see the references below). Therefore, it is important to understand the mechanisms that can explain the relationships between psychological distress and behaviors.
Alijanzadeh M, Harati T. The role of social capital in the implementation of social distancing during the COVID-19 pandemic. Asian J Soc Health Behav 2021;4:45-6
Ashraf A, Ali I, Ullah F. Domestic and gender-Based violence: Pakistan scenario amidst COVID-19. Asian J Soc Health Behav 2021;4:47-50
Rieger MO. Willingness to vaccinate against COVID-19 might be systematically underestimated. Asian J Soc Health Behav 2021;4:81-3
Shirali GA, Rahimi Z, Araban M, Mohammadi MJ, Cheraghian B. Social-distancing compliance among pedestrians in Ahvaz, South-West Iran during the Covid-19 pandemic. Asian J Soc Health Behav 2021;4:131-6
3. At the end of the Introduction, the authors should clearly mention the review purpose and what the specific review can address to the literature gap.
4. I am not sure whether the authors should keep the Methodology section as it reads like that the authors did not do a systematic review.
5. If the authors did a narrative review, I would suggest the authors restructure their manuscript. They should firstly describe different types of defense mechanisms. Then, describe the defense mechanisms under COVID-19 pandemic. It seems that the authors have mixed the defense mechanism findings before the pandemic with those after the pandemic. This hinders the readability.

Experimental design

The authors claimed that this is a review article; however, it is unclear what type of review design it is. The authors have searched PubMed using keywords. However, the authors did not provide details regarding the search process and findings (e.g., how many papers have been found and how the authors screen the papers). It seems like a narrative review from all the sections; however, the Methodology section lets me think that the authors might conduct something like systematic review. However, the structures do not look like systematic review. Therefore, the study design is confusing.

Validity of the findings

The authors did not report any findings. They mentioned that they have conducted a review search; however, they did not mention what the findings are. Instead, they directly describe narrative things.

Additional comments

Nil.

Reviewer 2 ·

Basic reporting

This is, in summary, an interesting manuscript aimed to provide a theoretical explanation of the psychological defense mechanisms used to deal with the pandemic then discussing different categories of defense mechanisms and their roles in managing the impacts of psychological distress. The authors found that evidence of the role of adaptive defense mechanisms in protecting people from psychological distress due to stressful life events, such as the COVID-19 pandemic was reported.
Here, as follows, my main comments/suggestions.

First, as the authors, throughout the Introduction section, correctly reported that Covid-19 is associated with defense mechanisms which are significant predictors of several variables, including personality traits, identity status, and psychopathology, they might further stress the link between psychological distress related to covid-11 and negative clinical outcomes such as suicidal behavior. Importantly, despite the continuous advancement in neuroscience as well as in the knowledge of human behaviors pathophysiology, currently suicidal behavior represents a puzzling challenge. Thus, given the importance of this topic, i suggest to cite within the main text the article published in 2018 on Int J Mol Sci (PMID: 30249029).

Also, the authors might immediately present and discuss, in the first lines of the Discussion section, their most relevant study findings. Conversely they seem to focus redundantly on the most relevant aims/objectives of this study that should have been stressed elsewhere within the main text.

Importantly, the most relevant study limitations/shortcomings should be appropriately described as the the main caveats have been only partially reported.

Finally, what is the take-home message of this manuscript? While the authors reported that the COVID-19 pandemic has had a wide range of negative physical and mental impacts psychosocial interventions play a critical role in reducing the risk of developing psychiatric and personality disorders due to a crisis or pandemic, they should, in my opinion, provide the most relevant conclusive remarks of their paper more extensively. Specifically, what are the main implications of the present findings? Here, more information may be useful for the general readership.

Experimental design

The most relevant aims/objectives of this study should be provided for general readership not in the methodology but in the Introduction section.

Importantly, why the present search has been conducted only on PubMed and not in other scientific databases (e.g., Scopus, ScienceDirect, Web of Sciences, etc) needs to be extensively explained by the authors.

Validity of the findings

The central sections are long and difficult to follow for the readers; thus, the introduction of one or more Tables/Figures in order to summarize the main text could be useful.

Additional comments

No additional comments.

---

## Round 0.2 · Minor Revisions

Manuscript is significantly improved by the authors. However, there are still some minor concerns raised by the reviewer. Please address these concerns and resubmit accordingly.

Reviewer 1 ·

Basic reporting

The authors have improved the readability of the manuscript.

Experimental design

I would like the authors to do the following:
1. Line 132. Change the sentence " This article has two emphases of interest:" into " This article serves as a narrative review has two emphases of interest:"
2. Move Lines 132-138 "This article has two emphases of interest: (1) the assessment of the negative influences of the COVID-19 pandemic on the psychological well-being and how the psychological distress, due to the pandemic, would affect and manipulate human’s psychological behavior; and (2) the importance of investigating different psychological defense mechanisms that play a critical role in mitigating psychological distress and utilizing mechanisms that promotes mental resilience, in addition, the evaluation of how adjustment mechanisms and coping strategies can protect peoples from internal and external threat." to the end of the Introduction (i.e., after the sentence of "Individuals’ responses to stressful situations might fall under one of the four categories of defense mechanisms; these responses can improve resilience and protect the ego from internal conflicts")
3. Line 127. Change "Survey methodology" into "Review methodology".

Validity of the findings

There is no problem here.

Additional comments

Nil.

Reviewer 2 ·

Basic reporting

In the revised manuscript, the authors addressed most of the major comments raised by Reviewers improving both the main structure and quality of the present paper.

Experimental design

No specific comments.

Validity of the findings

No specific comments.

Additional comments

No further additional comments/suggestions.

---

## Round 0.3 · accepted · Accept

The manuscript now can be accepted in its current form.